# Inoculation with *Actinobacteria* spp. Isolated from a Hyper-Arid Environment Enhances Tolerance to Salinity in Lettuce Plants (*Lactuca sativa* L.)

**DOI:** 10.3390/plants12102018

**Published:** 2023-05-18

**Authors:** Felipe González, Christian Santander, Antonieta Ruiz, Rodrigo Pérez, Jorge Moreira, Gladys Vidal, Ricardo Aroca, Cledir Santos, Pablo Cornejo

**Affiliations:** 1Doctorado en Ciencias Mención Biología Celular y Molecular Aplicada, Universidad de La Frontera, P.O. Box 54-D, Temuco 4780000, Chile; f.gonzalez31@ufromail.cl; 2Departamento de Ciencias Químicas y Recursos Naturales, Universidad de La Frontera, P.O. Box 54-D, Temuco 4780000, Chile; maria.ruiz@ufrontera.cl (A.R.); rodrigo.esteban.perez@gmail.com (R.P.); jorgem282@gmail.com (J.M.); cledir.santos@ufrontera.cl (C.S.); 3Grupo de Ingeniería Ambiental y Biotecnología, Facultad de Ciencias Ambientales y Centro EULA-Chile, Universidad de Concepción, Concepción 4070411, Chile; glvidal@udec.cl; 4Departamento de Microbiología del Suelo y la Planta, Estación Experimental del Zaidín, CSIC, Profesor Albareda 1, 18008 Granada, Spain; ricardo.aroca@eez.csic.es; 5Escuela de Agronomía, Facultad de Ciencias Agronómicas y de los Alimentos, Pontificia Universidad Católica de Valparaíso, Quillota 2260000, Chile

**Keywords:** Atacama Desert, *Metharme lanata*, saline water, plant-growth-promoting bacteria, phenolic compounds

## Abstract

Irrigated agriculture is responsible for a third of global agricultural production, but the overuse of water resources and intensification of farming practices threaten its sustainability. The use of saline water in irrigation has become an alternative in areas subjected to frequent drought, but this practice affects plant growth due to osmotic impact and excess of ions. Plant-growth-promoting rhizobacteria (PGPR) can mitigate the negative impacts of salinity and other abiotic factors on crop yields. *Actinobacteria* from the hyper-arid Atacama Desert could increase the plant tolerance to salinity, allowing their use as biofertilizers for lettuce crops using waters with high salt contents. In this work, rhizosphere samples of halophytic *Metharme lanata* were obtained from Atacama Desert, and actinobacteria were isolated and identified by 16S gene sequencing. The PGPR activities of phosphate solubilization, nitrogen fixation, and the production of siderophore and auxin were assessed at increasing concentrations of NaCl, as well as the enhancement of salt tolerance in lettuce plants irrigated with 100 mM of NaCl. Photosynthesis activity and chlorophyll content, proline content, lipid peroxidation, cation and P concentration, and the identification and quantification of phenolic compounds were assessed. The strains *S. niveoruber* ATMLC132021 and *S. lienomycini* ATMLC122021 were positive for nitrogen fixation and P solubilization activities and produced auxin up to 200 mM NaCl. In lettuce plants, both strains were able to improve salt stress tolerance by increasing proline contents, carotenoids, chlorophyll, water use efficiency (WUE), stomatal conductance (gs), and net photosynthesis (A), concomitantly with the overproduction of the phenolic compound dicaffeoylquinic acid. All these traits were positively correlated with the biomass production under saltwater irrigation, suggesting its possible use as bioinoculants for the agriculture in areas where the water resources are scarce and usually with high salt concentrations.

## 1. Introduction

Irrigated agriculture accounts for 33% of global agricultural production, using 25% of arable lands and consuming about 65% of available fresh water [1]. Irrigation in agriculture provides stability and food security, promoting increased production of both seasonal crops such as corn and vegetables, as well as permanent crops like small and large fruit trees. Additionally, it has optimized the use of agricultural soils [2]. Nevertheless, the overexploitation of hydric resources and agricultural intensification, such as double or triple cropping, determines a threat to the sustainability of agroecosystems and food production [3]. In numerous irrigated areas, farmers are being forced to use saline waters in irrigation due to limitations in the supply of waters of sufficient quality for irrigation together with an increase in the demand from other water users [4].

Nonetheless, the utilization of water containing elevated salinity levels for irrigation purposes may hinder the growth of plants due to the osmotic impact, which diminishes the plant’s capacity to absorb water, and the surplus of ions that affects the plant’s cells metabolism [5]. The diminished growth of plants due to salinity is caused by the toxicity of specific ions, such as Na^+^, that substitute nutrients like K^+^, Ca^2+^, and Mg^2+^ in the rhizosphere, rendering them inaccessible to plants [6].

The quality of irrigation waters is determined by the quantity and type of salts it contains, being the higher salinity of water used for irrigation one of the main sources of soil salinization and the concomitant crop yield reductions [7]. Salinity can have an impact in various physiological and metabolic processes, potentially limiting crop production depending on its severity and extent [8]. In the early stages of plant growth, salinity leads to a physiological drought because of high solute concentrations in the soil. This reduces the roots’ ability to absorb water and causes leaf dehydration. Additionally, salinity leads to ionic stress, where an accumulation of Na^+^ causes toxicity, ionic imbalances, and K^+^ deficiencies [9]. The above properties subsequently cause an increase in reactive oxygen species (ROS) production, causing oxidative damage and serious alterations in enzyme and macromolecule structure. This damage impacts cell tissues and alters photosynthesis, respiration, and protein synthesis [10]. To face oxidative damage generating by overproduction of ROS under stress conditions, plants have enzymatic and non-enzymatic antioxidant resources. The phenolic compounds are secondary metabolites which participate in the protection against ROS in the non-enzymatic way, and they include flavonoids, tannins and hydroxycinnamate esters. The modulation of ROS by phenolic compounds is due to their ability to be an electron or hydrogen donor, to stabilize the unpaired electron, and to chelate transition metal ions [11]. The synthesis of these molecules in plants can be modulated by the association with PGPR microorganisms and improve the antioxidant capacity by non-enzymatic means [12,13].

Several studies have been conducted to address the salinity stress, using various approaches such as the exploration of new techniques in conventional agriculture, developing novel plant varieties, utilizing genetic engineering, and incorporating bioinoculants. These efforts aim to mitigate the negative impacts of abiotic factors on crop yields [14]. The use of bioinoculants has resulted an eco-friendly biological alternative, which has obtained widespread acceptance, especially the utilization of PGPR. These bacteria have considerable potential for utilization in agriculture as they enhance crop productivity and protect plants against diverse types of abiotic stresses [15,16]. In recent years, there has been an increase in the utilization of actinobacteria in agricultural practices, owing to their potential as plant growth-promoting (PGP) and their widespread distribution in plants [17]. *Actinobacteria* employ various strategies to overcome abiotic stress, such as the synthesis of osmolytes, plant hormones, and enzymes; the regulation of the osmotic balance; and the increase of nutrient availability. Due to these qualities, actinobacteria are highly regarded as potential microbial inoculants [18].

The Atacama Desert in the extreme north of Chile is known for its extreme conditions. It is the most arid non-polar ecosystem in the world, while also including high salinity levels and, therefore, has been identified as a potential source of extremophile microorganisms that may have noticeable salt tolerance mechanisms. These microorganisms could have potential characteristics as plant-growth promoters when they are inoculated in crops under salinity conditions [19]. However, there are few studies on the interactive effects of irrigation with saline water and the bioinoculation of lettuce plants with different actinobacterial species; none of those included isolates from the hyper-arid Atacama Desert, giving a high level of novelty based in the use of unique biological resources in agriculture. Lettuce (*Lactuca sativa* L.) is a widely consumed fresh vegetable that is sensitive to salinity, with a threshold electrical conductivity of 1.3 dS m^−1^ [20]. Based on the above, the present study deals with the hypothesis that the species of actinobacteria isolated from the Atacama Desert have the ability to improve the tolerance of lettuce plants to salt stress by modulating antioxidant compounds and reducing oxidative damage. In this way, the aim of this study has been to investigate the potential of native strains of actinobacteria that were isolated from the Atacama Desert able to display PGP traits, and to demonstrate their efficacy as biofertilizers for lettuce crops. This study sought to minimize the reduction in lettuce yield associated with the stress induced by irrigating with saline water.

## 2. Results

### 2.1. Actinobacteria Identification

Four actinobacterial strains were isolated from different sites in the hyper-arid Atacama Desert, which were named as ATMLC32021, ATMLC132021, ATMLC122021, and ATMLC222021. For each strain, the 16S rRNA gene was amplified and sequenced, followed by submission to a BLASTN analysis to search for similarities. The sequences of the actinobacterial strains, together with their closest relatives in the GenBank database, were used to construct a phylogenetic tree using the neighbor-joining method, and the sequence of the genus *Micromonospora* was used as outgroup (Figure 1). Two distinct clades were obtained in the phylogenetic analyses, corresponding to the genera *Pseudarthrobacter* and *Streptomyces*. The isolate ATMLC32021 was grouped into the *Pseudarthrobacter* clade and showed a 98.99% sequence similarity with *Pseudarthrobacter siccitolerans* strain E111 (MF681807.1). The isolates ATMLC132021, ATMLC122021, and ATMLC222021 were grouped into the Streptomyces clade and showed sequence similarities of 99.93%, 99.08%, and 99.79% with *S. niveoruber* strain 173640 (EU570418.1), *S. lienomycini* strain FG.S.392 (KF991646.1), and *S. ambofaciens* strain F3 (KT368940.1), respectively. After the identification of the actinobacterial strains, the sequences were submitted to the GenBank database under the accession numbers OQ402204.1, OQ402202.1, OQ402203.1, and OQ402201.1 for the isolates ATMLC32021, ATMLC132021, ATMLC122021, and ATMLC222021, respectively.

### 2.2. Plant Growth Promotion Traits

The plant growth promotion traits were tested in all actinobacterial strains. Clear halo zones around their bacterial colonies grown on Pikovskaya’s media were observed for *P. siccitolerans* ATMLC32021, *S. ambofaciens* ATMLC222021, and *S. niveoruber* ATMLC132021, indicating a positive P solubilization. Siderophore production was positive for *S. niveoruber* ATMLC132021 and *S. lienomycini* ATMLC122021, as indicated by the presence of clear halo zones around their bacterial colonies grown on a CAS agar plate. In the case of N fixation, all strains tested showed a blue halo around the colonies grown on an NfB medium, indicating a positive capacity for N fixing (Table 1).

The auxin production was observed in the four strains grown on ISP2 supplemented with L-Tryptophan and at all the levels of NaCl analyzed (Table 2). The highest concentration (12.13 ± 0.26 µg mL^−1^) was produced by *P. siccitolerans* ATMLC32022 at 100 mM NaCl, and the lowest (6.46 ± 0.61 µg mL^−1^) by *S. lienomycini* ATMLC122021 at 200 mM NaCl. Furthermore, no significant differences were observed among the strains at 0 and 100 mM NaCl. However, when the strains were cultured with 200 mM NaCl, a significant decrease in the production of IAA was observed in *S. niveoruber* ATMLC132021 and *S. lienomycini* ATMLC122021.

### 2.3. Biomass Production

The production of shoot and root biomass, as well as the STI, were strongly influenced by salinity levels and their interactions with the inoculation of the different strains (*p* < 0.0001; Figure 2; Appendix A). Under salinity irrigation, all treatments had lower biomass production, with the lowest shoot fresh weight obtained in the plants inoculated with *P. siccitolerans* ATMLC32022 (PSC3) (17.41 ± 0.38 g) and *S. lienomycini* ATMLC122021 (SLC12) (7.29 ± 0.38 g) for the root biomass. A significant increase in shoot production was observed in plants inoculated with *S. lienomycini* ATMLC122021 (SLC12) (23.38 ± 0.53 g) and *S. niveoruber* ATMLC132021 (SNC13), (20.41 ± 0.09 g), with an increase of 23% and 5% compared to non-inoculated plants (19.05 ± 0.34 g). The production of root biomass, using irrigation with 100 mM NaCl, was improved only with the inoculation of *P. siccitolerans* ATMLC32022 (17.41 ± 0.38 g), with no differences in the remaining treatments compared to non-inoculated control (7.60 ± 0.42 g). In accordance with the biomass production, the salt tolerance index (STI) in plants inoculated with *S. lienomycini* ATMLC122021 reached the highest value at 100 mM NaCl, compared to all other treatments.

### 2.4. Photosynthetic Traits and Pigments

Photosynthesis (A; Figure 3A) and stomatal conductance (gs; Figure 3D) were strongly influenced by salt stress (*p* < 0.0001), as shown in all plants irrigated with 100 mM NaCl, which had lower values compared to plants treated with 0 mM NaCl. For net photosynthesis (A), no difference was observed between inoculation types in both levels of salinity analyzed, while stomatal conductance (gs) improved in plants inoculated with *S. lienomycini* ATMLC122021 (SLC12) and *P. siccitolerans* ATMLC32022 (PSC3) under 0 mM and 100 mM NaCl, respectively. The transpiration rate (E; Figure 3B) and water use efficiency (WUE; Figure 3C) were influenced by salinity irrigation, inoculation type, and the interaction of both variables (*p* < 0.05). Under salinity stress, WUE improved in plants inoculated with *S. ambofaciens* ATMLC222021 (SAC22), *S. niveoruber* ATMLC132021 (SNC13), and the highest value was obtained by *S. niveoruber* ATMLC132021 (SNC13). For transpiration rate in non-stress condition, the plants inoculated with *P. siccitolerans* ATMLC32022 (PSC3) and *S. ambofaciens* ATMLC222021 (SAC22) increased their transpiration rate. However, under salt stress conditions, only in the treatment in which plants were inoculated with *P. siccitolerans* ATMLC32022 (PSC3) were higher values in comparison with control plants seen, and no observed differences were observed in the remaining treatments.

The chlorophyll A, B, total chlorophyll, and carotenoids were strongly influenced by inoculation, NaCl, and the interaction of all factors (ANOVA *p* < 0.0001) (Figure 4). Under salinity stress conditions, chlorophyll A, B, and carotenoids were positively affected by the inoculation of *S. lienomycini* ATMLC122021 (SLC12). On the contrary, they were negatively affected by the inoculation of *P. siccitolerans* ATMLC32022 (PSC3) and *S. ambofaciens* ATMLC222021 (SAC22). Under non-saline stress conditions, no statistically significant differences were observed between treatments. However, the chlorophyll B content increased in plants inoculated by *S. ambofaciens* ATMLC222021 (SAC22).

### 2.5. Cations and Phosphorus Uptake

In the shoot Na^+^, K^+^, and in the root K^+^ concentrations were strongly influenced by the inoculation, salinity, and the interaction of both factors (*p* < 0.001). Nevertheless, the Na^+^ concentration in roots and the Na^+^/K^+^ ratio in both tissues were only affected by the NaCl factor (ANOVA *p* < 0.0001) (Table 3). The concentration of Na^+^ in leaves and roots was higher in all plants irrigated with 100 mM NaCl compared to plants not subjected to saline stress. In this way, the concentration of Na^+^ in leaves was decreased by the inoculation of *P. siccitolerans* ATMLC32022 (PSC3), *S. niveoruber* ATMLC132021 (SNC13), and *S. lienomycini* ATMLC122021 (SLC12). In roots, the Na^+^ concentration was only increased in plants inoculated with *P. siccitolerans* ATMLC32022 (PSC3). In the case of K^+^, the inoculation of *P. siccitolerans* ATMLC32022 (PSC3) and *S. ambofaciens* ATMLC222021 (SAC22) decreased its concentration in leaves and increased in root tissues, compared with control plants. The Na^+^/K^+^ ratios of both leaves and roots were higher in all treatments under 100 mM NaCl. However, in leaves, the ratio decreased in comparison to the control 100 mM NaCl treatment plant due to the inoculation of *P. siccitolerans* ATMLC32022 (PSC3), *S. niveoruber* ATMLC132021 (SNC13), and *S. lienomycini* ATMLC122021 (SLC12). In contrast, in the root tissue, the Na^+^/K^+^ ratio was decreased only by the inoculation of *S. lienomycini* ATMLC122021 (SLC12).

In leaves, P was affected by inoculation and salt factors, but not by the interaction of these two variables (ANOVA *p* < 0.001). While root P was only affected by the type of inoculation (ANOVA *p* < 0.01) (see Appendix A). Thus, leaf P was only statistically increased by the inoculation of *P. siccitolerans* ATMLC32022 (PSC3) under non-salt stress conditions; for root P, a decrease was observed in plants irrigated with 100 mM NaCl and inoculated with *S. lienomycini* ATMLC122021 (SLC12). The Mg^+^ concentration in leaves and root tissue was strongly influenced by NaCl factors (*p* < 0.0001) (Appendix A). However, in leaves, the Mg^+^ concentration was affected by the interaction of both factors. Under salinity stress, the Mg^+^ concentration in leaves was negatively affected by the inoculation of *P. siccitolerans* ATMLC32022 (PSC3); in roots, no statistical difference was observed between treatments. The accumulation of Ca^+^ in leaves was affected by the interaction of NaCl and inoculation factors (ANOVA *p* < 0.001), and in roots by the inoculation factor alone (ANOVA *p* < 0.05) (see Appendix A). No differences were observed in the accumulation of Ca^+^ in leaves and root tissues under salinity stress. However, under non-saline conditions, an increase in the accumulation of Ca^+^ in leaves was observed with the inoculation of *S. ambofaciens* ATMLC222021 (SAC22) and *S. niveoruber* ATMLC132021 (SNC13). In the root tissue, an increase in the accumulation of Ca^+^ was observed with the inoculation of *S. niveoruber* ATMLC132021 (SNC13) and *S. lienomycini* ATMLC122021 (SLC12).

### 2.6. Oxidative Damage and Proline Production

The production of proline was strongly affected by the factors NaCl and the interaction with inoculation type (ANOVA *p* < 0.001) (Figure 5). The proline content showed higher values in all plants irrigated with 100 mM NaCl compared to plants growing under non-salt stress conditions. Conversely, the inoculation of *S. lienomycini* ATMLC122021 (SLC12) showed an improvement in the production of proline at 100 mM NaCl. The oxidative damage measured by thiobarbituric acid reactive substances (TBARS) was affected by the inoculation type, salinity levels, and the interaction of these two factors (ANOVA *p* < 0.001). Under non-salt stress conditions, no statistically significant differences among treatments were observed. However, under salinity irrigation, a decrease in lipid peroxidation was observed in the plants inoculated with *P. siccitolerans* ATMLC32022 (PSC3), *S. niveoruber* ATMLC132021 (SNC13). The highest oxidative damage was reached in the treatment of plants inoculated with *S. ambofaciens* ATMLC222021 (SAC22).

### 2.7. Identification and Quantification of Phenolic Compounds and Antioxidant Capacity

A total of nine different phenolic compounds were identified through HPLC-DAD-ESI-MS/MS (Table 4, Appendix A). However, only seven compounds can be quantified by HPLC-DAD, and their concentrations were determined by comparing them with standards of chlorogenic acid and quercetin-3-glucoside. This was measured through the absorbance at wavelengths of 320 nm and 360 nm, respectively (Figure 6). The phenolic compounds 5-caffeoylquinic acid, chicoric acid, dicaffeoylquinic acid, and non-identified compounds with a retention time of 9.00 (RT9.00) had a greater concentration in leaves, in all treatments compared to non-salinity stress conditions. Under salinity stress, the phenolic compounds 5-caffeoylquinic acid and RT9.00 were greater increased in plants inoculated with *P. siccitolerans* ATMLC32022 (PSC3), and the compounds chicoric acid and dicaffeoylquinic acid in the plants inoculated with *S. niveoruber* ATMLC132021 (SNC13) and *S. lienomycini* ATMLC122021 (SLC12), respectively. The phenolic compounds quercetin-hexoside and quercetin-acetylhexoside had lower concentrations in contrast with the rest of the phenolic compounds identified. In the case of quercetin-hexoside, the higher values were observed in the control plants and inoculated with *S. lienomycini* ATMLC122021 (SLC12) under non-salinity stress conditions. While subjected to saline stress, a decrease in the value of the compound quercetin-hexoside was observed in the plants inoculated with *S. lienomycini* ATMLC122021 (SLC12), and it was not observed in the plants inoculated with *S. ambofaciens* ATMLC222021 (SAC22). Instead, the compound quercetin-acetylhexoside showed no differences in contrast to the control plants under salinity and non-salinity stress conditions. Although the phenolic compounds were increased differently depending on the type of inoculation, there were no significant statistical differences in the total phenolic content and antioxidant capacity, as determined by DPPH, TEAC, and CUPRAC (see Appendix A).

### 2.8. Multivariate Analysis

A principal component analysis was performed (PCA) separately by salinity levels to discover which traits are more associated with each treatment (Figure 7). For the plants growing in a non-salt stress condition, the two first principal components (PCs) explained 37.73% of the cumulative variance observed, with 23.82% on the first axis and 13.19% on the second. The most contributing variables in a positive way for PC1 were the phenolic compounds coumaroylquinic acid, 5-caffeoylquinic acid, dicaffeoylquinic acid and quercetin-hexoside, TEAC, and DPPH. Negatively contributing variables were the root and leaves’ K^+^ concentration (K_R and K_L, respectively), and Ca^2+^ leaf concentrations. The most contributing positive variables in PC1 were associated with control treatments and plants inoculated with *S. lienomycini* ATMLC122021 (SLC12). While the negative variables contributing on PC1 were associated with the treatment of inoculated by *P. siccitolerans* ATMLC32022 (PSC3), *S. ambofaciens* ATMLC222021 (SAC22), and *S. niveoruber* ATMLC132021 (SNC13). Under salt stress, the two first principal components (PCs) explained 43.31% of the cumulative variance observed, with 25.22% on the first axis and 18.09% on the second. The main variables that contributed positively in PC1 were FWS, total chlorophyll (ChlTT), chlorophyll (A), chlorophyll (B), and proline content; leaves’ Ca^2+^, and Mg^2+^; and the phenolic compound dicaffeoylquinic acid. These variables were associated with plant-inoculated *S. lienomycini* ATMLC122021 (SLC12), while the variables that were negatively correlated to PC1 were the phenolic compounds coumaroylquinic acid and chicoric acid, FWR, leaves P, and phenolics and were associated with plants inoculated with *P. siccitolerans* ATMLC32022 (PSC3) and *S. ambofaciens* ATMLC222021 (SAC22).

## 3. Discussion

Actinobacteria have been proposed as a good alternative to use as a bioinoculant due to minimal nutritional needs, easy propagation, storage, and a wide spectrum of solubilization mechanisms. In fact, they have the capacity to produce mycelium and dry spores, which allows them to adhere to soil surfaces and remain in a dormant state for longer periods, conferring resistance to extreme environments [21]. The Atacama Desert has been a source of actinobacterial strains with different biotechnological applications, including new bioactive metabolites and a range of plant growth-promoting abilities [22]. These characteristics of desert actinobacteria have been used to improve salt tolerance in crops [23,24]. In this work, we insolated four actinobacterial strains from the Atacama Desert and based on 16S gene sequencing one insolated had 98.99% of similarity with the genus *Pseudarthrobacter* and the rest of them had 99.08–99.79% with the genus *Streptomyces.* Both genera isolated here have been reported, such as plant-promoting rhizobacteria with the ability of fixing nitrogen, solubilizing phosphate, producing siderophores and auxin, and was use for improving salt tolerance in tomato [25,26], sage [27], maize [28], wheat [29], and soybean [30].

Auxins are phytohormone-derived, which helps in the growing and developing. They contribute to the developing of roots, root hair, and root laterals and have been produced by different halotolerant bacterial [31,32]. In our case, all actinobacterial strains can produce auxin in different concentrations of the NaCl tested, in a range of 6.46–12.13 µg mL^−1^, with a decrease in the strain *S. niveoruber* ATMLC132021 and *S. lienomycini* ATMLC122021 (Table 2). The present findings are consistent with prior research studies, which reported that actinobacteria produce less than 35 μg mL^−1^ of auxins, as evidenced by the observed outcome [33,34].

The ability to convert atmospheric N_2_ to ammonia was tested in an NfB medium. All strains isolated from the Atacama Desert was capability to fixing atmospheric N_2_. This trait was observed in endophytic and free-living rhizoactinobacteria, such as the genera *Arthrobacter*, *Agromyces*, *Corynebacterium*, *Myco-bacterium*, *Micromonospora*, *Propionibacteria*, and *Streptomyces*, which have nitrogen-fixing capacity [35]. Phosphate is an essential nutrient for the growth and development of plants. In this sense, the use of PGPR, which help in the solubilization of phosphate, makes it available for absorption by the roots and, thus, improves the plant’s ability to withstand abiotic stress [36]. In this sense, the capacity of actinobacteria to solubilization phosphate was reported in *Streptomyces*, *Nocardiopsis* [21], *Dermococus* [37], *Actinobacter*, and *Pseudarthrobacter* [28]. This report was in concordance with our findings, in which the phosphate solubilization was positive for insolate *P. siccitolerans* ATMLC32022, *S. ambofaciens* ATMLC222021 and *S. niveoruber* ATMLC132021.

Another important nutrient for plants is iron. In the soil, iron its abundant in ferric form (Fe^+3^), which is difficult for plants assimilate in the roots. Siderophores are metabolites that chelate iron and allow plants to easily absorb it through the roots [38]. PGPR can produce siderophores, which make iron available for plant absorption. The production of siderophores in actinobacteria has been mainly reported in the genera *Streptomyces* and *Amycolatopsis* sp. [39]. In our study, we observed the production of siderophore in the *S. niveoruber* ATMLC132021.

It is well known that salt stress reduces crop yield by restricting plant photosynthesis and biomass accumulation [6]. This is consistent with our research that has demonstrated that salinity has a significant impact on lettuce plant performance, resulting in reduced biomass production for both inoculated and non-inoculated plants. Nonetheless, the plants inoculated with actinobacterial species *S. niveoruber* ATMLC132021 and *S. lienomycini* ATMLC122021 showed an improvement in the biomass production of lettuce plants. *Actinobacteria* are one of the major components of rhizosphere microbial populations, playing a significant ecological role in soil nutrient cycling and plant-growth-promoting activities [18]. Numerous reports are available on their potential as agents for promoting plant growth [40,41]. Our results are in concordance with Mohamad et al. (2022), who reported a significant increase in the total fresh weight of cotton plants under different salt conditions when the plants were inoculated with actinobacterial strains *Streptomyces luteus* XIEG05 and *Nocardiopsis dassonvillei* XIEG12 [42]. In the same way, the *Streptomyces* sp. PGPA39 mitigated salt stress and promoted the growth of tomato plants that were subjected to salinity stress [26]. In our case, the plants that were inoculated with *S. lienomycini* ATMLC122021 achieved a higher value on the salt tolerance index (STI)—the index that has a positive and significant correlation with the potential yield under non-stressed or stressed conditions and determines the most tolerant plants to salinity stress conditions [43].

Plants exposed to salt stress can suffer from various physiological problems related to their metabolism, water balance, and mineral nutrition. Potassium is an essential nutrient for plants, playing a vital role in many physiological processes, including water regulation, photosynthesis, and protein synthesis [44]. In contrast, sodium is not crucial for plant growth: its high concentrations in plant tissues can lead to water stress due to how it interferes with water uptake and transport. Additionally, high sodium levels can lead to ion toxicity, disrupting various biochemical and metabolic processes in the plants and triggering oxidative stress [45]. In our study, the plants inoculated with *S. niveoruber* ATMLC132021 and *S. lienomycini* ATMLC122021 showed a lower Na^+^ concentration in leaves compared to control plants at 100 mM NaCl. According to research, some types of transporters in plants are responsible for removing Na^+^ from the xylem flow, thereby reducing its transport and accumulation in the shoots. This mechanism aids in preventing Na^+^ toxicity in the shoots by enabling the recirculation of Na^+^ back to the roots [46]. The HKT transporter family, which includes high-affinity K^+^ transporters, is essential for plant mechanisms that enable tolerance to Na^+^. Specifically, the family mediates Na^+^-specific transport or Na^+^-K^+^ co-transport and is responsible for regulating Na^+^ and K^+^ homeostasis, as well as preventing Na^+^ from reaching the shoot [47]. Due to a lower concentration of sodium in the leaves, the Na^+^/K^+^ ratio of lettuce plants was observed to decrease in our study, indicating a more favorable response to high levels of salt. The Na^+^/K^+^ ratio plays a crucial role in determining the ability of plants to tolerate saline stress [48]. Our results agree with what was determined by Moncada et al. (2020), who found the inoculation of the lettuce plants by bacteria with PGP characteristics significantly increased K^+^/Na^+^ ratio under various salinity levels which correlated with a high production of biomass [49].

Salt stress produces various physiological changes such as a decrease in photosynthesis rate, smaller stomatal aperture, lower stomatal conductance, decreased transpiration rate, and decreased chlorophyll concentration and fluorescence [50]. Photosynthetic activity is crucial for biomass productivity, which is strongly affected by salinity [51], and stomatal limitations are widely recognized as the primary factor contributing to reduced photosynthesis during mild-to-moderate stress [52]. We mainly found a reduction in stomatal conductance (gs) and net photosynthesis (A) in all treatments subject to saline stress. Similar results were found in the lettuce plant (*Lactuca sativa* L.) cultivar ‘Cheong Chi Ma’ when were irrigated with saline water [53]. Moreover, in our study, when lettuce plants were irrigated with saline water and inoculated with *S. niveoruber* ATMLC132021 and *S. lienomycini* ATMLC122021, there was a tendency for an increase in the gs and A parameters which could be directly related to an increase in biomass production in the plants. Regarding water use efficiency (WUE), we observed an improvement in this physiological factor when lettuce plants were watered with saline water. Furthermore, the inoculation with *S. ambofaciens* ATMLC222021, *S. niveoruber* ATMLC132021, and *S. lienomycini* ATMLC122021 resulted in higher WUE compared to the control group and their respective treatments without saline. The influence of salt levels on WUE has been reported in two tomato cultivars [54] and citrus plants [55] under varying levels of salinity. In two lettuce cultivars, similar results were found by Oliveira et al. (2021), who determined that an increase in WUE under salt stress can be interpreted as a mechanism of tolerance to NaCl in lettuce plants [56]. Under salt stress, the transpiration rate decreased by the stomatal closure reducing water lost through transpiration [57]. However, in our study, there was no observed difference in the plants with greater values of fresh biomass compared with control plants under 100 mM of NaCl irrigation. Thus, these plants can reach better growth and development with the same water expense.

Chlorophylls and carotenoids are important plant pigments that play crucial roles in photosynthesis and photoprotection. These pigments can change in response to different stressors, such as salinity and drought, which can affect the plant’s ability to adapt and perform photosynthesis [58]. When there is an excess of Na^+^ concentration in the cytosol, Na^+^ takes over the K^+^ binding sites, which results in chlorophyll degradation and interferes with the optimal functioning of proteins [48]. It has been determined that the chlorophyll a and b content, as well as the total chlorophyll content, decreased in two wheat cultivars (Gonbad and Zarin) that were irrigated with a saline solution containing 100 mM of NaCl; on the contrary, an increased amount of carotenoid content in Zarin cultivar was found [29]. In our study, we observed an increase in chlorophyll a and b, as well as total chlorophyll content, in lettuce plants that were inoculated with *S. lienomycini* ATMLC122021 and irrigated with saline water. Our results are in accordance with the findings in tomato plants inoculated with *Streptomyces* sp. PGPA39 [26], in rice plants inoculated with *Streptomyces* sp. GMKU 336 [59], and *Sorghum bicolor* inoculated with *Streptomyces* sp. RA04 and *Nocardiopsis* sp. RA07 [60].

As mentioned earlier, when plants are exposed to salt stress, their initial response is drought stress caused by osmotic shock, followed by the induction of stomatal closure. This, in turn, limits the plants’ photosynthetic capacity by restricting the supply of CO_2_ [61]. The fixation of CO_2_ and thylakoid reactions of photosynthesis occur within the thylakoids and stroma of the chloroplast, which supply vital carbon for growth, energy to drive various metabolic reactions, and the synthesis of diverse metabolites [62]. Chloroplasts are major producers of reactive oxygen species (ROS) via both photosystems due to the surplus of photons trapped in photosystem II (PSII) and the electron transfer to molecular oxygen via photosystem I (PSI) [63]. ROS substantially disrupt the regular metabolic processes by oxidizing lipids, nucleic acids, and proteins, leading to the destruction of proteins and peroxidation of membrane lipids [64]. Based on our findings, the control treatment and plants that were inoculated with *S. ambofaciens* ATMLC222021 exhibited higher levels of lipid peroxidation in their leaves. Conversely, plants that were inoculated with *P. siccitolerans* ATMLC32022 (PSC3), *S. niveoruber* ATMLC132021 (SNC13), and *S. lienomycini* ATMLC122021 showed a decrease in lipid peroxidation compared to the control plants. Plants have antioxidant mechanisms, enzyme systems, and phenolic compound production to mitigate the harmful effects of ROS overproduction caused by salt stress. These mechanisms scavenge free radicals and reduce oxidative damage, while also functioning as osmoprotectants and signaling molecules to maintain proper cytoplasmic osmotic balance [20]. Similarly, plants produce multiple osmolytes or osmoprotectants, such as proline, in their cytosol to aid in osmotic adjustment and protect against oxidative damage. Proline plays an essential role as an osmoregulator in plants by reducing cytoplasmic osmotic potential, promoting water absorption, and scavenging ROS molecules [65]. The present research demonstrated an enhancement in proline production when lettuce was inoculated with *S. niveoruber* ATMLC132021 (SNC13) and *S. lienomycini* ATMLC122021 under saline water irrigation. This result was positively correlated to a reduction in lipid peroxidation, as measured by TBARs, and an increase in leaf biomass production.

As previously stated, phenolic compounds are an important group of secondary metabolites produced by plants that can play a crucial role in protecting against ROS [66]. The present research has detected nine distinct phenolic compounds in lettuce plants, which is consistent with previous studies that have also identified the presence of nine phenolic compounds in lettuce plants subjected to salt stress conditions [67]. The expression of phenolic compounds in lettuce depends on the genotype, salinity levels, and can be modulated by the presence of the PGPR microorganism [20,68,69]. In the current study, the phenolic compound dicaffeoylquinic acid was linked to the inoculation of *S. lienomycini* ATMLC122021. This compound has been reported in lettuce plants acting with a high ROS scavenging activity [70,71]. In this sense, our results are in line with reported work on lettuce, where phenolic compounds are overexpressed under 100 mM of NaCl irrigation [69]. However, Santander et al. (2020) and Ayuso-Calles et al. (2020) found that lettuce plants produced higher levels of phenolic compounds, which were mainly associated with increased lipid peroxidation levels and reduced biomass production [72,73]. In this sense, to the best of our knowledge, this is the first report of overproduction of dicaffeoylquinic acid in lettuce plants in association with the inoculation of *S. lienomycini* ATMLC122021 was related to an improvement in salt stress tolerance in lettuce plants. The overproduction of phenolic compounds, enhanced by the actinobacteria strains, can deal with the ROS products generated by salt stress, decreasing lipid peroxidation, and consequently increasing biomass production.

In Figure 7B of the multivariable analysis, it was observed that the treatment linked to high biomass production exhibited a correlation with the overproduction of proline, phenolic compounds (such as dicaffeoylquinic acid), increased WUE, high chlorophyll content, and elevated carotenoid content. The high production of proline resulting from the inoculation of lettuce plants with *S. lienomycini* ATMLC122021 (SLC12) and *S. niveoruber* ATMLC132021 (SNC13) appears to serve as a primary response to counteract the physiological drought caused by reduced water potential in the soil from irrigation with saline water [67]. Additionally, proline serves a secondary function by protecting vital macromolecules from oxidative damage by improving plant performance [65], thus promoting the synthesis of essential phenolic compounds such as dicaffeoylquinic acid and photosynthetic pigments. This accumulation ultimately enables the maintenance of consistent photosystem levels while using less water (high WUE). In the same way, the inoculation with these actinomycetes also leads to an improvement in the Na^+^/K^+^ ratio in roots and shoots through K^+^ accumulation in roots and Na^+^ translocation from leaves to roots, enhancing the ionic balance and, thereby, increasing salt stress tolerance.

## 4. Materials and Methods

### 4.1. Isolation of Actinobacteria

The actinobacteria species were isolated from the rhizosphere of an endemic plant of Chile belonging to the Zygophyllaceae family, *Metharme lanata* (Phill), in different sites of the Atacama Desert (20°50′58,732″ S, 69°11′16,541″ W Quebrada de Cahuiza; 21°6′32,022″ S, 69°6′31,05″ W; 21°6′32,127″ S, 69°6′30,391″ W; 21°6′38,387″ S, 69°8′48,066″ W Quebrada de Choja, Tarapacá Region, Northern Chile). The soils were sampled at 5–20 cm depth, collected aseptically using a metallic shovel disinfected with 70% ethanol, and stored in sterile plastic bags. Samples were kept at 4 °C in a cooler and immediately transported to the laboratory for processing. A soil suspension was prepared by adding 1 g of soil to 49 mL of sterile NaCl 0.9% solution followed by a 10-fold serial dilution. *Actinobacteria* were isolated by inoculating 100 µL of each dilution on an ISP2 agar medium (yeast extract 4.0 g, malt extract 10.0 g, dextrose 4.0 g, agar 20.0 g, distilled water 1000 mL, pH = 7.2) and then incubating the plates at 30 °C for five days. After purification by repeated streaking on ISP2 agar medium, different colonies were selected based on their morphological characteristics such as shape, color, and size. The purified actinobacterial strains were stored at −80 °C in a 50% glycerol solution [74].

### 4.2. Molecular Identification of Actinobacteria

Total genomic DNA was extracted from 2 mL of a 7-day-old samples of each actinobacterial strain cultured on liquid ISP2 broth medium, using a Wizard^®^ Genomic DNA Purification Kit (Promega, Madison, WI, USA) according to the manufacturer’s instructions. Polymerase chain reaction (PCR) was performed for amplification of the 16S ribosomal DNA (rDNA) gene using GoTaq^®^ DNA Polymerase kit (Promega, Madison, WI, USA) in 25 μL per reaction with a final concentration of 1× colorless buffer, 3 mM MgCl_2_, 1 mM dNTPs, 0.4 μM of each universal primer for 16S ribosomal DNA sequence amplification (27F: 5′-AGAGTTTGATCCTGGCTCAG-3′ and 1492R: 5′-GGTTACCTTGTTACGACTT-3′) [75,76], and 1.25 U per reaction of GoTaq^®^ DNA Polymerase and approximately 0.25 μg of genomic DNA. The amplification conditions were as follows: 94 °C for 5 min for initial DNA denaturation, 35 cycles at 94 °C for 30 s, 60 °C for 30 s, 72 °C for 30 s, and a final elongation step at 72 °C for 5 min. The amplified products were analyzed by gel electrophoresis. The PCR products were sequenced from both directions using the 16S rDNA Forward and Reverse primers on an automated DNA sequencer ABI PRISM 3500xL (Applied Biosystems, CA, USA) by the sequencing service of Pontificia Universidad Católica, Santiago, Chile (CONICYT-FONDEQUIP EQM150077).

The sequences were assembled and manually edited in BioEdit v.7.2.5. The resulting nucleotide sequences were analyzed using the BLAST program [77] and compared to the NCBI GenBank database. The closely related sequence and actinobacterial sequence were aligned using CLUSTALW and then manually adjusted for phylogenetic analysis. The phylogenetic tree was constructed using the neighbor-joining method [78] by applying the Kimura-2-Parameter model [79] in the MEGA-X software version 10.2.3 [80]. The confidence value of the node was supported by bootstrap analyses based on 1000 bootstrap replications [81]. The sequence of the actinobacterial samples was submitted to the GenBank database.

### 4.3. Characterization of Plant Growth Promotion Traits

To determine siderophore production, phosphate (P) solubilization, and nitrogen (N) fixation, each actinobacterial strain was cultured in liquid ISP2 media and incubated at 30 °C and 120 rpm for 7 days. After incubation, the culture was disrupted by vortexing for 5 min, and OD600 was adjusted to 0.8. The potential to produce siderophores was assessed by inoculating 10 µL of the adjusted culture on a Chrome Azurol S (CAS) agar plate [82] and incubating for 7 days at 30 °C in the dark. Positive siderophore production was indicated by a color change (yellow, orange, purple, or purplish-red) in the zone around the actinobacterial colony. Screening for P solubilizing activity was determined by inoculating 10 μL of the adjusted actinobacterial culture in the center of a Pikovskaya’s agar (PVK) plate with 0.5% (*w*/*v*) tri-calcium phosphate as the sole P source [83] and incubating for 7 days at 30 °C in the dark. A clear zone of tri-calcium phosphate solubilization around the actinobacterial colony indicates positive P solubilization. Nitrogen fixation was tested by inoculating 10 µL of the adjusted actinobacterial culture in an NfB semisolid medium [84] and incubating for 7 days at 30 °C in the dark. Positive N fixation was observed in the presence of a turbid ring on the subsurface of the NfB medium. Three replicates were made for each strain for each analysis.

The auxin production was evaluated under different conditions of salinity as described by Rangseekaew et al. (2021) [37]. Each actinobacterial strain was cultured in glucose–yeast extract broth (glucose 10 g L^−1^, yeast extract 10 g L^−1^) (GYE) and incubated at 30 °C and 120 rpm for 7 days. After incubation, the culture was disrupted by vortexing for 5 min and the OD600 was adjusted to 0.8. GYE supplemented with 2 mg mL^−1^ of L-tryptophan (Sigma-Aldrich, Beijing, China) and different concentrations of NaCl were added with 100 µL of the adjusted actinobacterial culture and incubated at 30 °C and 120 rpm in the dark. The NaCl concentrations evaluated corresponded to 0, 100, and 200 mM. The supernatant was collected by centrifugation at 11,000 rpm for 15 min and evaluated for auxin production using the colorimetric assay by mixing 100 μL of the supernatant with 200 µL of Salkowski’s reagent [85]. After 30 min of incubation at room temperature in the dark, auxin production was detected spectrophotometrically by measuring the absorbance at 530 nm. The quantity of auxin production was determined from a standard curve using <99.9% pure IAA (Sigma-Aldrich, Beijing, China).

### 4.4. Actinobacterial Inoculum Preparation

Each actinobacterial strain was cultured in liquid ISP2 media and incubated at 30 °C and 120 rpm for 7 days. After incubation, the culture was disrupted by vortexing for 5 min, centrifuged at 5000 rpm for 5 min, and the supernatant was discarded. The bacterial pellet was resuspended in 5 mM MgSO_4_ and the OD600 was set to 0.8.

### 4.5. Experimental Design and Biological Material

A fully randomized factorial 5 × 2 design was used. The first experimental factor was inoculation (IN) comprising the following levels: non-inoculated plants (control), plants inoculated with *Pseudarthrobacter siccitolerans* ATMLC32021 (PSC3), plants inoculated with *Streptomyces ambofaciens* ATMLC222021 (SAC22), plants inoculated with *Streptomyces niveoruber* ATMLC132021 (SNC13), and plants inoculated with *Streptomyces lienomycini* ATMLC122021 (SLC12). The second experimental factor was irrigation with two saline levels: (i) non-saline conditions and (ii) plants irrigated with a solution of 100 mM NaCl. A total of 10 treatments, consisting of 4 replicates (n = 4; N = 40). The growing substrate consisted of a mixture of peat moss and perlite (70:30%, *v*:*v*), and the mix was autoclave-sterilized at 121 °C for 60 min on 3 consecutive days. Lettuce seeds (*Lactuca sativa* var. Gran Milanesa) were surface sterilized using 5% NaClO for 5 min and then washed twice with sterile distilled water to eliminate NaClO residues. The sterilized seeds were sown in polystyrene trays, and each cell of the tray was inoculated directly with 1 mL of the corresponding actinobacterial inoculum. After 20 days, the seedlings were transplanted to 1 L pots containing 300 g of the substrate mixture, and a second inoculation step was performed as above. At both times, non-inoculated plants received 1 mL of sterile 5 mM MgSO_4_.

### 4.6. Growth Conditions

The experiment was carried out for 45 days under greenhouse conditions (25/21 °C; 50/60% relative humidity, 14/10 h day/night photoperiod) at the Department of Chemical Sciences and Natural Resources, Universidad de la Frontera, Temuco, Chile. Salinity treatments were applied post-transplantation via irrigation with the respective saline or non-saline water. The plants were irrigated every 2 days with 50 mL of solutions containing 0 or 100 mM of NaCl (0.0004 and 10 dS m^−1^, respectively). An equal amount of tap water was applied on alternate days to maintain soil moisture near the field capacity and to prevent excessive salt accumulation.

### 4.7. Measurements in Plants

#### 4.7.1. Biomass Production, Cation, and P Uptake

At harvest, the shoot and root tissues were weighed, and subsamples (3 g) of fresh material were ground in liquid N_2_ to obtain a fine powder that was stored at −80 °C for subsequent analysis. The residual material was dried (65 °C, 48 h) in a forced-air oven for chemical analysis. Subsequently, 1 g of ground shoot and root dry material was pulverized, incinerated at 550 °C, and finally digested in an acid mixture of H_2_O/HCl/HNO_3_ (8/1/1, *v*/*v*/*v*). The digestates were used for the spectrophotometric determination of phosphorus (P) using the blue-molybdate method and determinations of potassium (K^+^), calcium (Ca^2+^), magnesium (Mg^2+^), and sodium (Na^+^) by atomic absorption spectroscopy (Unicam SOLAAR, mod. 969). The STI (salt tolerance index) for every treatment was calculated as the ratio of the total weight under salt treatment and the total weight of the control [86].

#### 4.7.2. Photosynthetic Traits and Pigments

The determination of transpiration rate (E: mmol H_2_O m^−2^ s^−1^), photosynthesis rate (A: μmol CO_2_ m^−2^ s^−1^), and stomatal conductance (gs: mmol H_2_O m^−2^ s^−1^) were carried out using the Targas-1 equipment (PP Systems, Amesbury, MA, USA) following the user’s manual instructions. Measurements were taken one day before harvest in the second-youngest leaf from four plants per treatment.

Photosynthetic pigments of four leaves per treatment were extracted using 0.5 g of fresh material in 5 mL methanol. Subsequently, samples were filtered through 13 mm diameter Millex filters with 0.22 um pore size nylon membrane (Millipore, Bedford, MA, USA). The absorbance of the supernatant was measured using a Synergy H1 Hybrid Multi-Mode microplate spectrophotometer (BioTek Inc., Winooski, VT, USA) at a wavelength of 663 nm and 645 nm for chlorophyll A and B, respectively. The concentrations of pigments were calculated according to the formula provided by Lichtenthaler et al., 1987 [87].

#### 4.7.3. Lipid Peroxidation and Proline Production

Lipid peroxides were extracted from 100 mg of freshly powdered leaf tissue using 1.5 mL of 0.2% trichloroacetic acid (TCA) in microtubes (2 mL). The mixture was vortexed for 10 seconds and centrifuged at 17,000× *g* for 10 min at 4 °C. The chromogen was formed by mixing 300 mL of the supernatant with 1.2 mL of a mixture containing 20% (*w*/*v*) TCA and 0.5% 2-thiobarbituric acid, being the final mixture incubated at 95 °C for 30 min. The microtubes were then rapidly cooled in an ice bath. The supernatants were used for spectrophotometric readings at 440 nm, 532 nm, and 600 nm using a Synergy H1 Hybrid Multi-Mode microplate spectrophotometer (BioTek Inc., Winooski, VT, USA). Malondialdehyde (MDA) contents were calculated as described by Du and Bramlage (1992) [88]. The free proline concentration was determined using 0.5 g of fresh tissue and spectrophotometrically assayed at 530 nm, as described by Bates (1973) [89].

#### 4.7.4. Identification and Quantification of Phenolic Compounds and Antioxidant Capacity

Phenolic compounds were extracted according to the methodology described by Aguilera et al. (2020), with minor modifications. First, 0.3 g of leaf samples were immersed in 5 mL of extraction solvent (methanol:formic acid, 95:5, *v*:*v*) [90]. The samples were subsequently sonicated with an ultrasonic processor at 130 W (Sonics and Materials, Connecticut, USA) for 60 s at 40% amplitude, shaken for 30 min at 200 rpm, and finally centrifuged for 10 min at 4000× *g*. The supernatant was then transferred to another tube, protected from light, and stored at −20 °C until the measurements of phenolic compounds and antioxidant activities were performed. For the chromatographic determinations, the above extract was filtered with 0.22 µm pore filters and injected into amber vials. High-performance liquid chromatography-diode array detection (HPLC-DAD) analyses were carried out using an HPLC system equipped with a quaternary LC-20AT pump with a DGU-20A5R degassing unit, a CTO-20A oven, a SIL-20A autosampler, and a UV-visible diode array spectrophotometer (SPD-M20A); (Shimadzu, Tokyo, Japan). Instrument control and data collection were carried out using Lab Solutions software (version 5.96) (Shimadzu, Duisburg, Germany). Identity assignments were performed using an HPLC-DAD system coupled to a 6545-quadrupole time-of-flight (Q-ToF) mass spectrometer (Agilent, Waldbronn, Germany). The control software used here was a Mass Hunter workstation (version B.06.11). The MS/MS conditions used were previously reported by Santander et al. (2022) [69]. Identities were assigned by comparing the MS/MS spectra with those from the literature data and commercial standards. The MS/MS conditions used were previously reported by Santander et al. (2022) [69].

Total phenol concentrations were determined using the Folin–Ciocalteu method, as described by Singleton and Rossi (1965) [91], and adapted for use with a microplate reader, as described in Parada et al. (2019) [92]. The absorbance was measured at 750 nm, and gallic acid was used as the standard. The results were expressed as mg of gallic acid per g of fresh weight (FW). Antioxidant activities were determined using the DPPH (2,2-diphenyl-1-picrylhydrazyl), TEAC (Trolox equivalent antioxidant activity), and CUPRAC (cupric ion reducing antioxidant activity) methods, as reported by Fritz et al. (2022) [93].

### 4.8. Statistical Analysis

All statistical analysis and figures were performed in R version 4.2.1. Two-way ANOVA was used to test for significant differences between measurements of each experimental variable. For the variables with significant differences, the means were compared using the Fisher LSD multiple range test with the package “agricolae” v. 1.3.5. Moreover, the dataset was split by salinity level and subjected to a principal component analysis (PCA) for each salinity level. Confidence ellipses (group means) by inoculation treatment were also generated using the packages “FactoMineR” v. 2.7 and “factoextra” v. 1.0.7.

## 5. Conclusions

In conclusion, actinobacterial strains were isolated for the first time from the Atacama Desert and were found to be associated with *Metharme lanata* plants. Four actinomyces were identified, but only *Streptomyces niveoruber* and *Streptomyces lienomycini* are reported for the first time as promoting plant growth and improving salt tolerance in lettuce. The strains isolated in this study showed PGPR activities such as fixing atmospheric nitrogen, solubilizing phosphate, producing siderophores, and Auxin at increasing concentrations of salt. Lettuce plants under salt stress in association with these actinobacteria can improve salt tolerance due to the overproduction of proline, the phenolic compound dicaffeoylquinic acid, increased WUE, improved photosynthetic performance, diminished Na^+^ concentration in leaves, and a diminishing Na^+^/K^+^ ratio which is correlated with high biomass production and lower lipid peroxidation. In these senses, actinobacteria isolated from the Atacama Desert have the potential to be used as bioinoculants for improving crop production and using saline irrigation water. Further research is necessary to explore the mechanisms of plant growth promotion by these bacteria and their potential application in agriculture.

## Figures and Tables

**Figure 1 plants-12-02018-f001:**
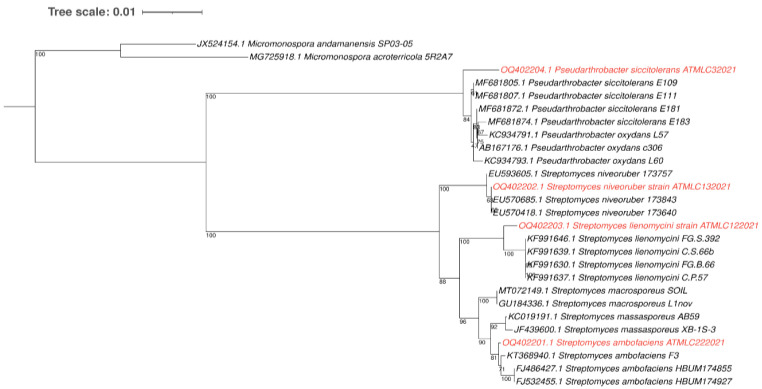
Phylogenetic tree obtained through the neighbor-joining method based on 16S rRNA gene sequence. The confidence value of the node was supported by bootstrap analyses based on 1000 replications and was represented at each node. Bars represent 0.01 substitutions per nucleotide position. The genus *Micromonospora* was used as an outgroup. The sequences obtained in this study was depicted in red color.

**Figure 2 plants-12-02018-f002:**
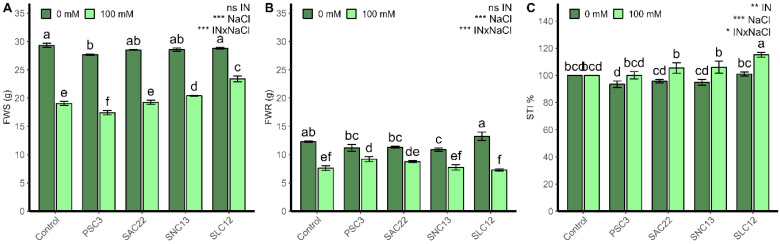
Fresh weight of shoots (FWS) (**A**), fresh weight of roots (FWR) (**B**), and salt tolerance index (STI) (**C**) of lettuce plants non-inoculated (Control) or inoculated with the actinobacteria *P. siccitolerans* ATMLC32022 (PSC3), *S. ambofaciens* ATMLC222021 (SAC22), *S. niveoruber* ATMLC132021 (SNC13), and *S. lienomycini* ATMLC122021 (SLC12) under salinity stress (100 mM NaCl, light green) and in the absence of salt (0 mM NaCl, dark green). The data include means ± SE (n = 4). The data were analyzed through a two-way ANOVA performed with inoculation and salinity stress as sources of variation. The significant difference was depicted as ns: no significant, *p* < 0.01: *, *p* < 0.001: **, and *p* < 0.0001: ***. Different letters indicate significant differences (*p* ≤ 0.05) according to Fisher’s multiple range test.

**Figure 3 plants-12-02018-f003:**
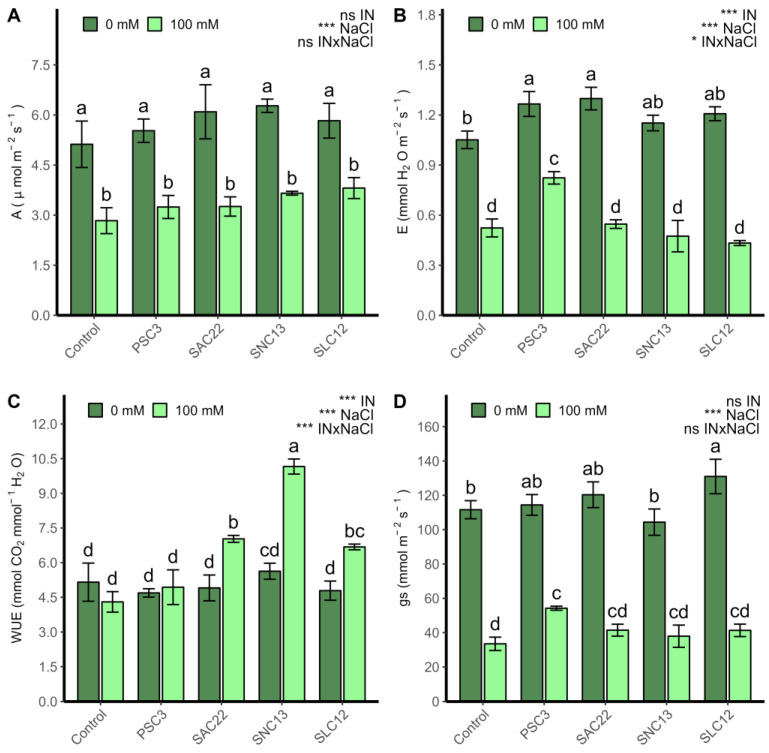
Photosynthesis (**A**), Transpiration rate (**B**), water use efficiency (**C**), and stomatal conductance (gs) (**D**) of lettuce plants non-inoculated (Control) or inoculated with the actinobacteria *P. siccitolerans ATMLC32022* (PSC3), *S. ambofaciens* ATMLC222021 (SAC22), *S. niveoruber* ATMLC132021 (SNC13), and *S. lienomycini* ATMLC122021 (SLC12) under salinity stress (100 mM NaCl, light green) and in the absence of salt (0 mM NaCl, dark green). The data include means ± SE (n = 4). The data were analyzed through a two-way ANOVA performed with inoculation and salinity stress as sources of variation. The significant difference was depicted as ns: no significant, *p* < 0.01: * and *p* < 0.0001: ***. Different letters indicate significant differences (*p* ≤ 0.05) according to Fisher’s multiple range test.

**Figure 4 plants-12-02018-f004:**
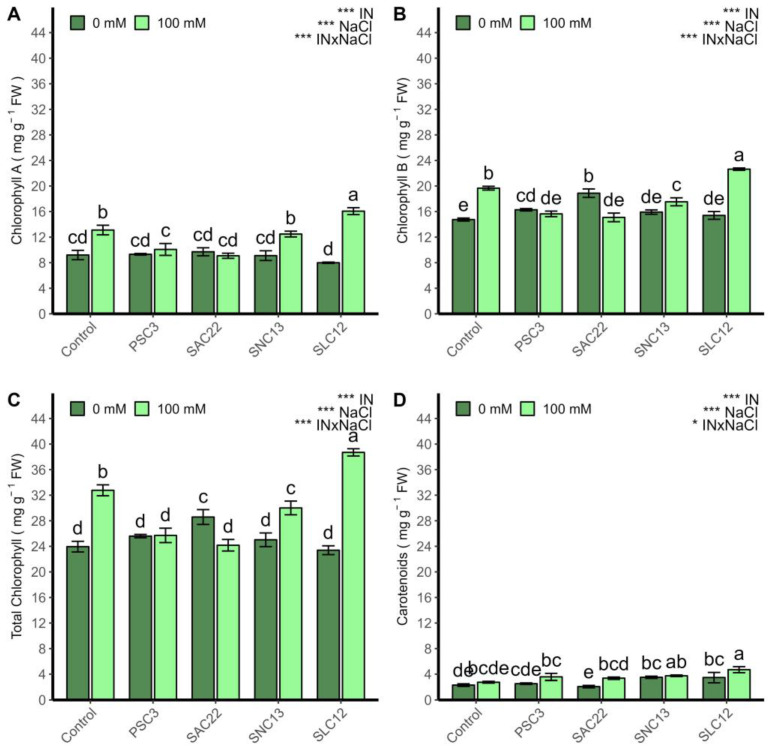
Chlorophyll A (**A**), chlorophyll B (**B**), total chlorophyll (**C**), and carotenoids (**D**) of lettuce plants non-inoculated (Control) or inoculated with the actinobacteria *P. siccitolerans* ATMLC32022 (PSC3), *S. ambofaciens* ATMLC222021 (SAC22), *S. niveoruber* ATMLC132021 (SNC13), and *S. lienomycini* ATMLC122021 (SLC12) under salinity stress (100 mM NaCl, light green) and in the absence of salt (0 mM NaCl, dark green). The data include means ± SE (n = 4). The data were analyzed through a two-way ANOVA performed with inoculation and salinity stress as sources of variation. The significant difference was depicted as *p* < 0.01: * and *p* < 0.0001: ***. Different letters indicate significant differences (*p* ≤ 0.05) according to Fisher’s multiple range test.

**Figure 5 plants-12-02018-f005:**
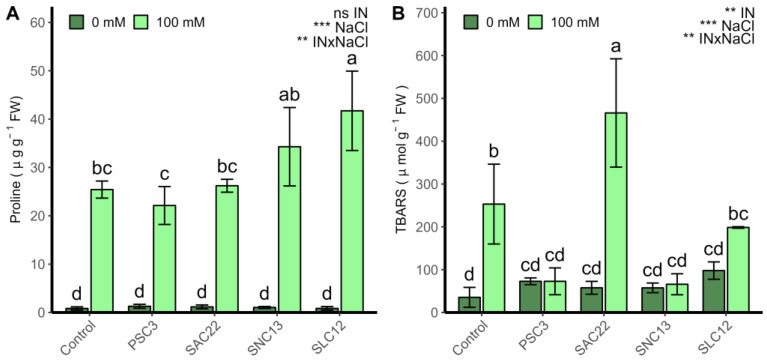
Proline content (**A**) and TBARS (**B**) in leaves of lettuce plants non-inoculated (Control) or inoculated with the actinobacteria *P. siccitolerans* ATMLC32022 (PSC3), *S. ambofaciens* ATMLC222021 (SAC22), *S. niveoruber* ATMLC132021 (SNC13), and *S. lienomycini* ATMLC122021 (SLC12) under salinity stress (100 mM NaCl, light green) and in the absence of salt (0 mM NaCl, dark green). The data include means ± SE (n = 4). The data were analyzed through a two-way ANOVA performed with inoculation and salinity stress as sources of variation. The significant difference was depicted as ns: no significant, *p* < 0.001: **, and *p* < 0.0001: ***. Different letters indicate significant differences (*p* ≤ 0.05) according to Fisher’s multiple range test.

**Figure 6 plants-12-02018-f006:**
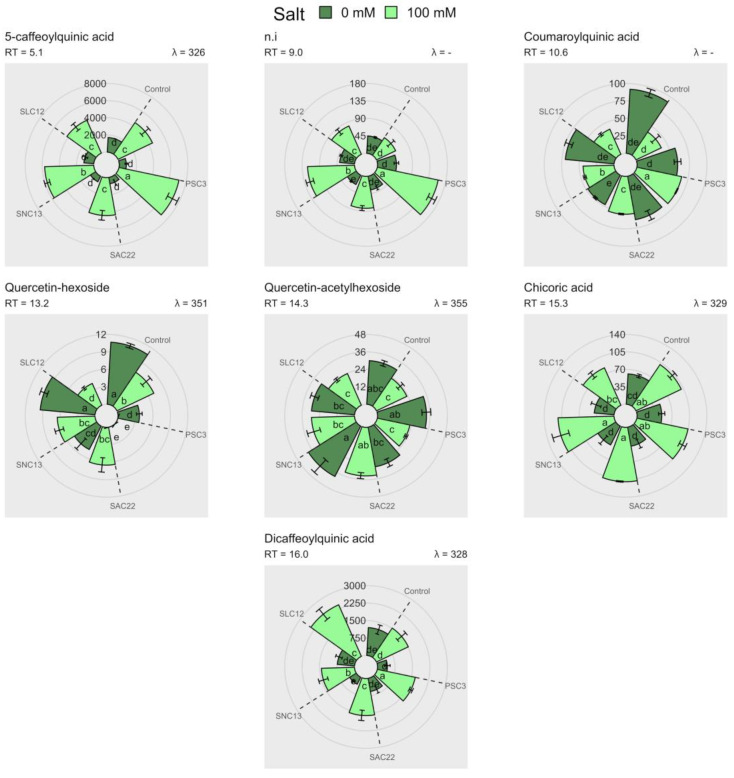
Phenolic compounds identified by HPLC-DAD in lettuce plants non-inoculated (control), inoculated with *P. siccitolerans* ATMLC32022 (PSC3), *S. ambofaciens* ATMLC222021 (SAC22), *S. niveoruber* ATMLC132021 (SNC13), and *S. lienomycini* ATMLC122021 (SLC12) under salinity stress (100 mM of NaCl in light green) and non-salt stress condition (0 mM of NaCl in dark green). The data include means ± SE (n = 4). The measurement of phenolics is computed and represented in mg g^−1^ FWR. The retention time (RT) and the wavelength (λ) used to measure the absorbance were also depicted. The data were analyzed through a two-way ANOVA performed with inoculation and salinity stress as sources of variation. Different letters indicate significant differences (*p* ≤ 0.05) according to Fisher’s multiple range test.

**Figure 7 plants-12-02018-f007:**
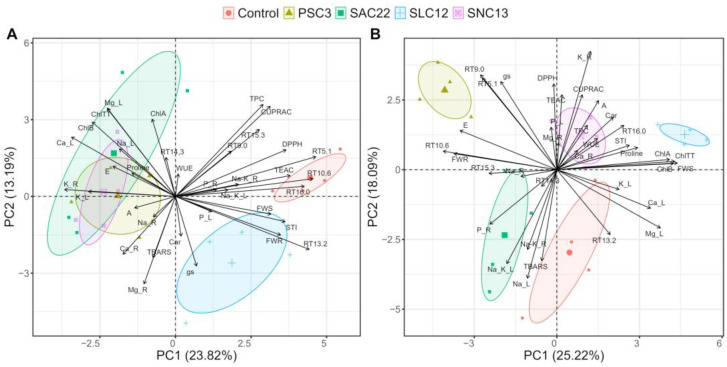
Principal Components Analysis (PCA) biplot for NaCl irrigation of 0 mM and (**A**) 100 mM of NaCl (**B**) based on biomass production of shoot (FWS) and root (FWR), photosynthetic parameters: net photosynthesis (A), transpiration rate (E), water use efficiency (WUE), and stomatal conductance (gs), photosynthetic pigments chlorophyll A (ChlA), chlorophyll B (ChlB), total chlorophyll (ChlTT), and carotenoids (Car), total phenolic compounds (TPC), antioxidant activities (AA) determined by DPPH, TEAC, and CUPRAC methods, oxidative damage determined by TBARS, proline content (proline), cations (Na^+^, Ca^2+^, Mg^2+^, and K^+^), and phosphorus uptake (P) in leaf (L) and root (R) tissues, and phenolic compounds determined by HPLC-DAD and MS/MS (RT5.1, RT9.0, RT10.6, RT13.2, RT14.5, RT15.3, and RT16) of lettuce plants non-inoculated (Control), inoculated by *P. siccitolerans* ATMLC32022 (PSC3), *S. ambofaciens* ATMLC222021 (SAC22), *S. niveoruber* ATMLC132021 (SNC13), and *S. lienomycini* ATMLC122021 (SLC12).

**Table 1 plants-12-02018-t001:** Production of siderophore, phosphate (P) solubilization and nitrogen (N) fixation of actinobacterial strain isolated from the hyper-arid Atacama Desert.

Strain	Siderophore	P Solubilization	N Fixation
*P. siccitolerans* ATMLC32021	−	+	+
*S. ambofaciens* ATMLC222021	−	+	+
*S. niveoruber* ATMLC132021	+	+	+
*S. lienomycini* ATMLC122021	+	−	+

“+”: Positive result; “−”: Negative result.

**Table 2 plants-12-02018-t002:** Auxin production (µg mL^−1^) by the different actinobacterial strains obtained from the Atacama Desert and growing at increasing NaCl concentrations.

Strain	0 mM NaCl	100 mM NaCl	200 mM NaCl
*P. siccitolerans* ATMLC32021	11.53 ± 0.23 ab	12.13 ± 0.26 a	10.75 ± 0.16 bc
*S. ambofaciens* ATMLC222021	11.35 ± 0.16 ab	11.27 ± 0.22 abc	11.4 ± 0.75 ab
*S. niveoruber* ATMLC132021	11.92 ± 0.30 ab	11.96 ± 0.13 ab	10.06 ± 0.11 c
*S. lienomycini* ATMLC122021	11.57 ± 0.90 ab	10.92 ± 0.26 abc	6.46 ± 0.61 d

Data represent the mean values of three replicates ± SD. Different letters indicate a significant difference in auxin production according to LSD fisher test at *p* < 0.05.

**Table 3 plants-12-02018-t003:** Na^+^ and K^+^ concentration in root and shoot of lettuce plants.

Treatments	Na^+^		K^+^		Na^+^/K^+^	
	Root (mg g^−1^)	Leaves (mg g^−1^)	Root (mg g^−1^)	Leaves (mg g^−1^)	Root	Leaves
**0 mM NaCl**						
Control	4.55 ± 0.71 c	2.20 ± 0.40 c	5.61 ± 1.02 d	12.88 ± 0.96 ef	0.95 ± 0.29 cd	0.18 ± 0.04 de
PSC3	7.45 ± 0.34 c	2.49 ± 0.20 c	8.63 ± 0.83 b	25.68 ± 1.28 ab	0.88 ± 0.08 cd	0.10 ± 0.01 e
SAC22	4.82 ± 0.58 c	2.49 ± 0.10 c	7.90 ± 1.19 bc	12.07 ± 0.68 f	0.70 ± 0.2 d	0.19 ± 0.04 de
SNC13	5.69 ± 0.82 c	3.08 ± 0.53 c	8.95 ± 0.09 b	28.09 ± 1.64 a	0.64 ± 0.09 d	0.11 ± 0.00 e
SLC12	6.45 ± 0.82 c	2.65 ± 0.22 c	6.51 ± 0.11 cd	10.69 ± 2.24 f	0.92 ± 0.16 cd	0.19 ± 0.05 de
**100 mM NaCl**						
Control	25.53 ± 0.41 ab	19.15 ± 0.83 a	8.49 ± 0.11 bc	26.19 ± 0.65 ab	2.56 ± 0.29 a	0.98 ± 0.24 b
PSC3	27.82 ± 2.06 a	12.31 ± 1.63 b	13.99 ± 0.66 a	17.89 ± 2.24 cd	1.98 ± 0.07 ab	0.69 ± 0.02 bc
SAC22	23.16 ± 3.3 b	20.39 ± 1.14 a	9.78 ± 1.13 b	16.68 ± 2 de	2.59 ± 0.64 a	1.30 ± 0.21 a
SNC13	21.48 ± 2.33 b	11.55 ± 0.32 b	13.11 ± 0.3 a	26.51 ± 0.28 a	2.14 ± 0.56 ab	0.44 ± 0.01 cd
SLC12	23.95 ± 1.33 ab	11.99 ± 0.96 b	15.01 ± 0.04 a	21.88 ± 1.67 bc	1.64 ± 0.09 bc	0.57 ± 0.08 c

Data represent the mean values of three replicates ± SD. Different letters indicate a significant difference between treatments in auxin production according to LSD fisher test at *p* < 0.05.

**Table 4 plants-12-02018-t004:** Identification of phenolic compounds by HPLC-DAD-ESI-MS/MS in lettuce leaves.

Peak	RT (min)	Compound	λ Max (nm)	[M − H]^−^	Product Ions
1	5.1	5-caffeoylquinic acid	326	353.1	191.1
2	9.0	n.i	-	431.2	295.0; 163.1
3	10.6	Coumaroylquinic acid	-	337.1	191.0
4	11.3	Caffeic acid	-	179.0	135.1
5	12.6	Quercetin-hexoside	-	463.1	301.0
6	13.2	Quercetin-hexoside	351	463.1	300.0
7	14.3	Quercetin-acetylhexoside	355	505.1	300.0
8	15.3	Chicoric acid	329	473.1	311.0; 149.0
9	16.0	Dicaffeoylquinic acid	328	515.1	353.1; 191.0

n.i: Not identified.

## Data Availability

The 16S ribosomal gene amplicon sequences have been deposited to the NCBI under the accession number OQ402204.1, OQ402202.1, OQ402203.1, and OQ402201.

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
