# Peer review of "Inoculation with Actinobacteria spp. Isolated from a Hyper-Arid Environment Enhances Tolerance to Salinity in Lettuce Plants (Lactuca sativa L.)"

_plants, 2023, doi:10.3390/plants12102018_

Round 1
Reviewer 1 Report
In the presented manuscript authors describes the characterization of four actinobacterial strains obtained from Atacama desert ant their possible effect as PGPR on lettuce plants irrigated with 100 mM NaCl. Authors performed large number of experiment to fully characterize the strains and there possible use as bioinoculants for improving crop production under salt stress.
Two of the strains (ATMLC132021 and ATMLC122021) were able to improve the biomass production under saltwater irrigation (from only 5 to up to 23% respectively).
Presented manuscript is well written and obtained results are intriguing but the quality of presentation can be further improved.
For example:
1. In the results part at the beginning of each new paragraph authors should try to explain shortly why one or other experiment was done. Authors performed extremely large number of different experiments but in some cases is not clear why exactly these experiments were done. Authors should clearly motivate the experimental conditions: why auxin production were study at 100 and 200mM NaCl (Tab.2) and plant tolerance only at 100 mM NaCl (Fig. 2). It was shown that even 40 or 80 mM NaCl affect the grow of lettuce plants.
2. In the discussion, instead of discussing their original data, authors are mainly looking for accordance of their single results with already published data. Authors should also try to discuss their results taken together. For example how they will explain the observed different Chlorophyll content (especially for plants inoculated with SLC12 and irrigated with 100mM NaCl, Fig. 4) but the same Photosynthesis rate (Fig. 3A)
3. How they will explain that the SLC12 is negative regarding P solubilization and have the lowest auxin production at 200mM NaCl but contribute to the highest shoot production under salt irrigation?
4. How authors will discuss the observed low antioxidant activities in inoculated plants non treated with NaCl in respect to non-inoculated plants?
Some minor points.
For me is surprising the extremely low standard error reported in Fig.2. In this case how the authors will explain much higher standard error reported in Fig.3 with material collected from the same plants?
In this type of articles including some photos of inoculated and non-inoculated plants threated with 0 or 100 mM NaCl will be a plus.
Row 189 “the highest WUE value was obtained by S. lienomycini ATMLC122021 (SLC12). In my copy SNC13 demonstrate the highest value.
Some of the references are cited by name, not by number (i.e row455: Oliveira et al.,; row 672: Aguilera et al.), please correct.
Row 605. Please describe better the composition of the nutrient solution used for irrigation.
The quality of English Language is good, although some sentences as “no observed differences was observed in the remaining treatments” (row 194) should be avoided.
Reviewer 2 Report
The manuscript described results of a greenhouse experiment on the effect of inoculation with Actinobacteria spp. on enhance tolerance to salinity in lettuce (Lactuca sativa L.). The subject is interesting and results provide an advance in current knowledge of possibility to use plant growth-promoting rhizobacteria (PGPR) to enhance tolerance to salinity in crop plants. I read the manuscript with attention. The methods are adequately described. The results are clearly presented. The conclusions are supported by the results. Overall the well written manuscript can be accepted for publishing in Plants after minor revision have been made.
Keywords should reflected the scientific content of the work but should not be repetition of the title words (Actinobacteria, Lactuca sativa). Please find such words (which are not be repetition of the title) that more detail reflecting the scientific content of the work, this way search engines of the web will be find your paper with higher probability. Please do not use abbreviations as a keywords (PGPR).
The hypothesis made in the study should be clearly stated.
Line 606: The first experimental factor was ….. Please clearly specify the second experimental factor. Please record the levels of the first experimental factor the same as the levels of irrigation (i), (ii) ….
